# COVID-19 vaccine effectiveness among South Asians in Canada

**Rahul Chanchlani**[1,2,3,4], **Baiju R. Shah**[2,5], **Shrikant I. Bangdiwala**[4,6], **Russell J. de Souza**[4,6], **Jin Luo**[2], **Shelly Bolotin**[7,8,9,10], **Dawn M. E. Bowdish**[11,12], **Dipika Desai**[3,4], **Karl Everett**[2], **Scott A. Lear**[13], **Mark Loeb**[4,14], **Zubin Punthakee**[4,12], **Diana Sherifali**[4,6], **Gita Wahi**[3,15], **Sonia S. Anand**[3,4,6,12]*

1 Division of Pediatric Nephrology, Department of Pediatrics, McMaster Children's Hospital, Hamilton, Ontario, Canada, 2 ICES, Toronto, Ontario, Canada, 3 Chanchlani Research Centre, McMaster University, Hamilton, Ontario, Canada, 4 Population Health Research Institute, McMaster University, Hamilton, Ontario, Canada, 5 Department of Medicine, University of Toronto, Toronto, Ontario, Canada, 6 Department of Health Research Methods, Evidence, and Impact, McMaster University, Hamilton, Ontario, Canada, 7 Centre for Vaccine Preventable Diseases, University of Toronto, Toronto, Ontario, Canada, 8 Dalla Lana School of Public Health, University of Toronto, Toronto, Ontario, Canada, 9 Department of Laboratory Medicine and Pathobiology University of Toronto, Toronto, Ontario, Canada, 10 Public Health Ontario, Toronto, Ontario, Canada, 11 Firestone Institute for Respiratory Health, St. Joseph's Healthcare, St. Joseph's Healthcare, Hamilton, Ontario, Canada, 12 Department of Medicine, McMaster University, Hamilton, Ontario, Canada, 13 Simon Fraser University, Burnaby, British Columbia, Canada, 14 Department of Pathology and Molecular Medicine, Hamilton, Ontario, Canada, 15 Department of Pediatrics, McMaster University, Hamilton, Ontario, Canada

* anands@mcmaster.ca

**Data Availability Statement:** The dataset from this study is held securely in coded form at ICES. Legal data sharing agreements between ICES and the data providers (including healthcare organizations and government) prohibit ICES from making the

## Abstract

We evaluated the effectiveness of COVID-19 vaccines among South Asians living in Ontario, Canada compared to non-South Asians and compared the odds of symptomatic COVID-19 infection and related hospitalizations and deaths among non-vaccinated South Asians and non-South Asians. This was a test negative design study conducted in Ontario, Canada between December 14, 2020 and November 15, 2021. All eligible individuals >18 years with symptoms of COVID-19 were subdivided by ethnicity (South Asian vs other) and vaccination status (vaccinated versus not). The primary outcome was vaccine effectiveness as defined by COVID-19 infections, hospitalizations, and deaths, and secondary outcome was the odds of COVID-19 infections, hospitalizations, and death comparing non-vaccinated South Asians to non-vaccinated non-South Asians. 883,155 individuals were included. Among South Asians, two doses of COVID-19 vaccine prevented 93.8% (95% CI 93.2, 94.4) of COVID-19 infections and 97.5% (95% CI 95.2, 98.6) of hospitalizations and deaths. Among non-South Asians, vaccines prevented 86.6% (CI 86.3, 86.9) of COVID-19 infections and 93.1% (CI 92.2, 93.8) of hospitalizations and deaths. Non-vaccinated South Asians had higher odds of symptomatic SARS-CoV-2 infection compared to non-vaccinated non-South Asians (OR 2.35, 95% CI 2.3, 2.4), regardless of their immigration status. COVID-19 vaccines are effective in preventing infections, hospitalizations and deaths among South Asians living in Canada. The observation that non-vaccinated South Asians have higher odds of symptomatic COVID-19 infection warrants further investigation.

dataset publicly available. However, access may be granted to those who meet pre-specified criteria for confidential access, available at www.ices.on.ca/DAS or das@ices.on.ca.

**Funding:** This study was financially supported by ICES, which is funded by the Ontario Ministry of Health and Long-Term Care, the COVID-19 Immunity Task Force, Public Health Agency of Canada in the form of an annual grant (2122-HQ-000058) received by to SSA for the Ontario Health Data Platform, a Province of Ontario initiative to support Ontario's ongoing response to COVID-19 and its related impacts. This study was also financially supported by Canada Research Chairs Program in the form of a Tier 1 Canada Research Chair in Ethnic Diversity and Cardiovascular Disease grant (#CRC-2017-00024) received by SSA. This study was also financially supported by Canada Research Chairs Program in the form of a Michael G DeGroote Heart and Stroke Foundation of Canada Chair in Population Health Research grant received by SSA. The funders had no role in study design, data collection and analysis, decision to publish, or preparation of the manuscript.

**Competing interests:** The authors have read the journal's policy and have the following competing interests: SSA holds a Tier 1 Canada Research Chair in Ethnic Diversity and Cardiovascular Disease (#CRC-2017-00024), and the Michael G DeGroote Heart and Stroke Foundation of Canada Chair in Population Health Research outside of the submitted work. ML sits on vaccine advisory boards for Seqirus, Pfizer, Sanofi, Medicago, GSK, Merck, Novovax, and Janssen; is on the Data Safety Monitoring Board for CanSino Biologics, has received funding from Seqirus for a vaccine trial, is receiving in-kind supply of smallpox vaccines from Bavarian Nordic, and has provided expert testimony about vaccines outside of the submitted work. DMEB holds a Tier 2 Canada Research Chair in Aging and Immunity, sits on vaccine advisory boards for Pfizer and AstraZeneca, has received consulting fees/honoraria for Pfizer and AstraZeneca, and has provided expert testimony about vaccines outside of the submitted work. SB is the Director of the Centre for Vaccine Preventable Diseases (CVPD) at the University of Toronto outside of the submitted work. The CVPD receives operational support from a mix of funding sources, including through donations from pharmaceutical companies. A robust set of governance practices are in place to safeguard the academic freedom of the CVPD. BRS is funded by the University of Toronto as the Novo Nordisk Research Chair in Equitable Care of Diabetes and Related Conditions outside of the submitted work.

# Introduction

Ethnic communities, which are vulnerable in terms of socioeconomic status and health inequality, faced heightened risk of COVID-19 infection related hospitalizations and mortality worldwide [1]. A recent population-based cohort study of 17 million adults from the United Kingdom found that South Asians, defined as people who originate from the Indian subcontinent, were at increased risk of testing positive for SARS-CoV-2 and of COVID-19-related hospitalizations, ICU admissions, and deaths [2].

South Asians are the largest non-white and fastest growing ethnic group in Canada. During the COVID-19 pandemic, indirect as well as direct assessment indicated that South Asians in the Greater Toronto Area of Ontario, Canada experienced a higher rate of COVID-19 compared to other ethnic groups [3].

Vaccines are effective in reducing the risk of COVID-19 infection and associated hospitalizations and deaths, as shown by Phase 3 randomized clinical trials. [4–7]. However, there is significant disparity in vaccine uptake among ethnic groups, especially South Asians [8–10]. Approximately 25% of people living in the Indian subcontinent are vaccine hesitant due to cultural or religious reasons, concomitant comorbidities, low health literacy, receipt of misinformation on social media, and due to the influence of their peers [11]. Furthermore, in COVID-19 vaccine trials, "Asian" participants in which South Asians would be included, were substantially underrepresented (<5%) [7, 12], and therefore, there are sparse data on effectiveness of COVID-19 vaccines among South Asians. These factors as well as timing and reason for immigration to Canada may limit vaccine confidence and therefore uptake amongst the South Asian population in Canada.

To address these concerns, we conducted a series of studies [13, 14], including this population-based study using health administrative databases among South Asians living in Ontario, Canada. The specific objectives of this analysis were to: 1) evaluate the effectiveness of COVID-19 vaccines among South Asians living in Ontario, Canada compared to non-South Asians, and 2) compare the odds of symptomatic COVID-19 infection and related hospitalizations and deaths among non-vaccinated South Asians and non-South Asians. Finally, we explored potential explanatory factors for differences in COVID-19 infection rates including co-morbid conditions, socioeconomic status, immigrant status, reason for and duration of immigration to Canada.

# Methods

This study is reported in accordance with the Reporting of studies Conducted using Observational Routinely collected health Data (RECORD) guidelines [15]. The Ontario Personal Health Information Protection Act states that the use of data in this project is authorized under section 45 and does not require to be reviewed by a research ethics committee or institutional board.

## Study population, setting and design

Using the methods described by Chung et al. [16], we conducted a population-based study with a test negative design in which we included all individuals 18 years and older living in Ontario who had symptoms consistent with COVID-19. Test-negative designs are often used for vaccine efficacy studies, and are a special case of a case-control study in which the controls are subjects undergoing the same tests for the same reasons, but who test negative instead of positive as cases [17]. All Ontarians who were tested by PCR for SARS-CoV-2 between December 14, 2020 and Nov 15, 2021, were eligible. We excluded those without OHIP (Ontario Health Insurance Plan) coverage (public health insurance) and those living in long-term care

DS holds the Heather M. Arthur Population Health Research Institute/Hamilton Health Sciences Chair in Inter-Professional Health Research and has received an honorarium from Diabetes Update 2023 for being an invited speaker and is a co-methods lead on the Diabetes Canada Clinical Practice Guidelines Steering Committee outside of the submitted work. GW has received grants from the Canadian Institutes of Health Research and the Hamilton Health Sciences Foundation outside of the submitted work. RJdS has received grants from the Canadian Institutes of Health Research, Canadian Foundation for Dietetic Research, Population Health Research Institute, and Hamilton Health Sciences Corporation; he has received consulting fees and travel honoraria from the World Health Organization's Nutrition Guidelines Advisory Group; he is an Independent Director for the Helderleigh Foundation (Canada) and Co-Chair of a method working group for the ADA/EASD Precision Medicine in Diabetes group outside of the submitted work. SIB has received grants from the Canadian Institutes of Health Research and from the International Development Research Centre; he also served as a member of the US National Institute of Allergy and Infectious Diseases Data and Safety Monitoring Board for COVID-19 Preventive monoclonal antibodies trials outside of the submitted work. All other authors declare that they have no competing interests. This does not alter our adherence to PLOS policies on sharing data and materials. There are no patents, products in development or marketed products associated with this research to declare.

homes. We restricted the analysis to individuals who had at least one relevant COVID-19 symptom at the time of testing [16].

Symptomatic individuals who tested positive at least once for SARS-CoV-2 were considered as cases. Individuals who were symptomatic but were negative on all tests for SARS-CoV-2 during the study period were considered as controls. For cases, the index date was the date of specimen collection for their positive test (or one selected at random, if they had multiple positive tests), and for controls, the date of a randomly selected negative test result was considered as the index date.

Further, this cohort was divided into four sub-samples based on ethnicity and vaccination status, as follows: South Asian vaccinated, South Asian non-vaccinated, non-South Asian vaccinated, and non-South Asian non-vaccinated. Only those who had received both vaccine doses before the PCR testing and were 7 or more days after the second dose were considered as vaccinated.

**Data sources.** We obtained information regarding COVID-19 vaccination status, including vaccine product, date of administration, and dose number, from COVaxON, a centralized COVID-19 vaccine information system in Ontario. Data on laboratory-confirmed SARS-CoV-2 infection was detected by real-time reverse transcription polymerase chain reaction (RT-PCR) collected from the Ontario Laboratories Information System (OLIS) for both individuals who tested positive and individuals who tested negative. OLIS also contains information on the symptoms which were collected from patients at the time of testing. We obtained information on the clinical course of cases from the Public Health Case and Contact Management system (CCM). Ethnicity was determined using the ETHNIC database, which uses a validated last name algorithm to classify individuals into South Asians and non-South Asian groups [18]. By design, this algorithm has high specificity for South Asian names, but low sensitivity. Immigration, Refugee and Citizenship Canada's Permanent Resident Database was used to determine the immigration status and duration. Demographic data were obtained from the Registered Persons Database (RPDB). Hospitalization data were obtained from the Discharge Abstract Database [using International Classification of Diseases, Ninth and Tenth Revision (ICD-9 and ICD-10) codes], and diagnostic and fee codes from physician billing claims were obtained from the OHIP database. These datasets were linked using unique encoded identifiers and analyzed at ICES (*formerly known as* Institute for Clinical Evaluative Sciences).

**Outcomes ascertainment.** The primary outcome was symptomatic SARS CoV-2 infection and COVID-19 associated hospitalizations and deaths among vaccinated South Asians and non-South Asians. COVID-19-related hospitalization was defined as a positive test result which occurred within 14 days before or three days after admission [19]. COVID-19-related death was identified as a positive test result which occurred within 30 days before death or within seven days postmortem [16].

**Secondary outcomes.** Included the odds of symptomatic SARS CoV-2 infection among non-vaccinated South Asians and non-South Asians; and the risk of COVID-19 associated hospitalization or death in COVID-19 infected non-vaccinated South Asians and non-vaccinated non-South Asians.

**Covariates.** Demographic variables included age, sex, neighborhood income quintile (by postal code), rural status (community <10,000 persons), immigration status, time since immigration and reason for immigration (economic, refugee and other). Pre-existing co-morbid conditions on ICES derived cohorts such as hypertension, diabetes, history of asthma, chronic obstructive pulmonary disease (COPD), cancer, CKD (chronic kidney disease), immunocompromised status, dementia/frailty, CHF (congestive heart failure), TIA/Stroke, cardiac ischemia, arrythmia were ascertained using various diagnostic and procedural codes (S1 Table).

### Statistical analysis

Chi-Square test was used to compare the 4 groups (South Asian vaccinated, South Asian non-vaccinated, non-South Asian vaccinated, and non-South Asian non-vaccinated). We used multivariable logistic regression models to calculate the odds ratio (and 95% confidence interval) for outcomes after adjusting for relevant covariates (age, sex, rural/urban, neighborhood income quintile and any comorbidity). We then calculated the vaccine effectiveness for both symptomatic SARS CoV2 infection and for COVID-19 related hospitalizations and deaths, using the following formula: (1 –odds ratio of the outcome among vaccinated versus non-vaccinated individuals x 100%), in both the South Asian and non-South Asian populations. We also compared the odds of COVID-19 infection and the risk of COVID-19 related hospitalization and deaths among South Asian non-vaccinated vs. non-South Asian non-vaccinated individuals. Goodness of fit of the models was assessed by the Hosmer-Lemeshow test. Stratified analyses were conducted using immigration status (non-immigrants, recent immigrants [<10 years] and non-recent immigrants [>/ = 10 years]), and, among immigrants only, the reasons for immigration (economic, refugee, and family/others).

We also performed two sensitivity analyses to confirm the robustness of our findings. First, we analysed COVID-19 related hospitalizations and death separately. Second, we excluded participants with pre-existing respiratory illnesses (as defined above). In addition, stratified analysis of vaccine effectiveness in South Asians and non-South Asians was performed based on the pandemic waves: Wave 2: December 14, 2020 (start of our data) to February 28, 2021; Wave 3: March 1, 2021 to July 31 2021 and Wave 4: August 1, 2021 to November 15, 2021 (end of our data) [20].

All analyses were conducted using SAS version 9.4 (SAS Institute, Cary, NC). Tests were two sided, with P<0.05 considered as statistically significant.

## Results

A total of 883,155 individuals who had a PCR test for SARS-CoV-2 between 14 December 2020 and November 15, 2021, were identified. Among them, 126,016 were cases and 757,139 were controls (S1 Fig). S2 Table depicts the cohort creation process. Of 344,411 participants who were excluded, 342,409 (99.4%) were <18 years of age. The cohort creation, baseline characteristics and comorbidities of the cases and controls are shown in S3–S5 Tables. The important SARS-CoV-2 variants during our study period were Alpha (B.1.1.7), Beta (B.1.351), Gamma (P.1), Delta (B.1.617.2), Omicron (B.1.1.529).

The overall cohort was divided into 4 subgroups as shown in Table 1. South Asians were younger, more frequently male, and less likely to live in rural communities compared to the non-South Asian group. More than half (54.8%) of South Asians resided in local health integrated networks 5 and 6, which included the cities of Mississauga and Brampton. Notably, South Asians had an overall lower prevalence of comorbid conditions compared to non-South Asians, except for diabetes mellitus (S2 Table). 54.7% of South Asians were immigrants compared to 17.6% of non-South Asians.

### Vaccine effectiveness

Among South Asians, 2 doses of COVID-19 vaccine were 93.8% (95% CI 93.2, 94.4) and 97.5% (95% CI 95.2, 98.6) effective in preventing SARS Co-V-2 infection and related hospitalizations or deaths, respectively. Among non-South Asians, vaccine effectiveness was lower at 86.6% (CI 86.3, 86.9) and 93.1% (CI 92.2, 93.8) for SARS Co-V-2 infection and related hospitalizations or deaths, respectively. (Table 2).

**Table 1. Baseline characteristics of the overall cohort stratified by ethnicity and vaccination status.**

| Characteristics | Value | South Asian vaccinated | South Asian non vaccinated | Non-South Asian vaccinated | Non-South Asian non vaccinated | P-VALUE |
|---|---|---|---|---|---|---|
| | | N = 12,281 | N = 28,595 | N = 262,162 | N = 580,117 | |
| Sex | F | 6,671 (54.3%) | 14,366 (50.2%) | 157,320 (60.0%) | 330,131 (56.9%) | <.001 |
| | M | 5,610 (45.7%) | 14,229 (49.8%) | 104,842 (40.0%) | 249,986 (43.1%) | |
| Age at index date | Mean ± SD | 41.05 ± 15.65 | 40.64 ± 15.16 | 45.26 ± 18.35 | 43.35 ± 17.40 | <.001 |
| | Median (IQR) | 38 (29–50) | 37 (29–50) | 42 (31–59) | 40 (29–56) | <.001 |
| Age group | 18–29 | 3,077 (25.1%) | 7,716 (27.0%) | 59,056 (22.5%) | 151,686 (26.1%) | <.001 |
| | 30–39 | 3,656 (29.8%) | 7,981 (27.9%) | 59,892 (22.8%) | 129,097 (22.3%) | |
| | 40–49 | 2,364 (19.2%) | 5,471 (19.1%) | 45,066 (17.2%) | 97,524 (16.8%) | |
| | 50–59 | 1,410 (11.5%) | 3,793 (13.3%) | 36,242 (13.8%) | 90,755 (15.6%) | |
| | 60–69 | 1,000 (8.1%) | 2,131 (7.5%) | 31,091 (11.9%) | 60,969 (10.5%) | |
| | 70–79 | 553 (4.5%) | 1,027 (3.6%) | 17,874 (6.8%) | 30,612 (5.3%) | |
| | 80+ | 221 (1.8%) | 476 (1.7%) | 12,941 (4.9%) | 19,474 (3.4%) | |
| Any comorbidity | | 5,183 (42.2%) | 11,797 (41.3%) | 126,632 (48.3%) | 270,918 (46.7%) | <0.001 |
| Rural | missing | 34 (0.3%) | 59 (0.2%) | 663 (0.3%) | 1,579 (0.3%) | <.001 |
| | No | 12,151 (98.9%) | 28,306 (99.0%) | 235,992 (90.0%) | 515,892 (88.9%) | |
| | Yes | 96 (0.8%) | 230 (0.8%) | 25,507 (9.7%) | 62,646 (10.8%) | |
| Income Quintile | missing | 34 (0.3%) | 59 (0.2%) | 791 (0.3%) | 1,894 (0.3%) | <.001 |
| | 1 | 1,533 (12.5%) | 4,063 (14.2%) | 42,137 (16.1%) | 114,907 (19.8%) | |
| | 2 | 2,576 (21.0%) | 6,830 (23.9%) | 48,654 (18.6%) | 115,472 (19.9%) | |
| | 3 | 3,337 (27.2%) | 8,256 (28.9%) | 51,976 (19.8%) | 115,547 (19.9%) | |
| | 4 | 2,685 (21.9%) | 5,611 (19.6%) | 56,938 (21.7%) | 116,860 (20.1%) | |
| | 5 | 2,116 (17.2%) | 3,776 (13.2%) | 61,666 (23.5%) | 115,437 (19.9%) | |
| Immigrant status | | 6,497 (52.9%) | 15,871 (55.5%) | 42.869 (16.4%) | 105,647 (18.2%) | <.001 |
| LHIN (Local Health Integrated Network) | 1 (Erie St. Clair) | 282 (2.3%) | 563 (2.0%) | 20,908 (8.0%) | 54,629 (9.4%) | <.001 |
| | 2 (South West) | 103 (0.8%) | 218 (0.8%) | 8,751 (3.3%) | 20,846 (3.6%) | |
| | 3 (Waterloo Wellington) | 629 (5.1%) | 1,435 (5.0%) | 23,469 (9.0%) | 46,228 (8.0%) | |
| | 4 (Hamilton Niagara Haldimand Brant) | 244 (2.0%) | 780 (2.7%) | 13,013 (5.0%) | 36,678 (6.3%) | |
| | 5 (Central West) | 3,886 (31.6%) | 11,375 (39.8%) | 13,592 (5.2%) | 34,357 (5.9%) | |
| | 6 (Mississauga Halton) | 2,317 (18.9%) | 5,141 (18.0%) | 27,362 (10.4%) | 72,033 (12.4%) | |
| | 7 (Toronto Central) | 681 (5.5%) | 1,228 (4.3%) | 25,888 (9.9%) | 48,742 (8.4%) | |
| | 8 (Central) | 1,620 (13.2%) | 3,075 (10.8%) | 34,878 (13.3%) | 69,135 (11.9%) | |
| | 9 (Central East) | 2,070 (16.9%) | 3,882 (13.6%) | 33,820 (12.9%) | 62,110 (10.7%) | |
| | 10 (South East) | 79 (0.6%) | 154 (0.5%) | 11,500 (4.4%) | 25,378 (4.4%) | |
| | 11 (Champlain) | 137 (1.1%) | 204 (0.7%) | 10,923 (4.2%) | 20,716 (3.6%) | |
| | 12 (North Simcoe Muskoka) | 147 (1.2%) | 364 (1.3%) | 16,966 (6.5%) | 36,502 (6.3%) | |
| | 13 (North East) | 52 (0.4%) | 87 (0.3%) | 12,050 (4.6%) | 29,300 (5.1%) | |
| | 14 (North West) | 34 (0.3%) | 89 (0.3%) | 9,042 (3.4%) | 23,463 (4.0%) | |
| COVID-19 test result | Indeterminate | 6 (0.0%) | 46 (0.2%) | 138 (0.1%) | 853 (0.1%) | <.001 |
| | Negative | 11,853 (96.5%) | 17,980 (62.9%) | 254,449 (97.1%) | 471,814 (81.3%) | |
| | Positive | 422 (3.4%) | 10,569 (37.0%) | 7,575 (2.9%) | 107,450 (18.5%) | |

**Table 2. Vaccine effectiveness among South Asians and non-South Asians.**

| Outcome | Effect | Odds Ratio | Lower CI | Upper CI | Vaccine effectiveness | Vaccine effectiveness lower CI | Vaccine effectiveness upper CI |
|---|---|---|---|---|---|---|---|
| Symptomatic covid19 infection | South Asian vaccinated vs South Asian non-vaccinated[1] (n = 40876) | 0.06 | 0.05 | 0.07 | 93.8 | 93.2 | 94.4 |
| | non-South-Asian vaccinated vs non-South-Asian non-vaccinated[1] (n = 842279) | 0.134 | 0.131 | 0.14 | 86.6 | 86.3 | 86.9 |
| Hospitalization or death associated with symptomatic COVID-19 infection | South Asian vaccinated vs South Asian non-vaccinated[2] (n = 30473) | 0.03 | 0.01 | 0.05 | 97.4 | 95.2 | 98.6 |
| | non-South-Asian vaccinated vs non-South-Asian non-vaccinated[1] (n = 735991) | 0.07 | 0.06 | 0.08 | 93.1 | 92.2 | 93.8 |

[1] Hosmer-Lemeshow test for goodness of fit p-value <0.05

[2] Hosmer-Lemeshow test for goodness of fit p-value >0.05

Results were similar when COVID-19 related hospitalizations and death were analysed separately (S6 Table). Vaccines had similar effectiveness among South Asians (95.5%, 95% CI 81.5, 98.9) to prevent COVID-19 associated deaths compared to non-South Asians (89.8%, 95% CI 87.2, 91.9). Vaccines effectiveness remained similar after excluding participants with pre-existing respiratory conditions (S7 Table).

## COVID-19 infection and related hospitalizations/deaths among non-vaccinated individuals

South Asian non-vaccinated individuals had two-fold higher odds of symptomatic COVID-19 infection (aOR 2.35, 95% CI 2.3, 2.4) compared to the non-South Asian non-vaccinated cohort (referent cohort), after adjustment for multiple potential covariates. Similarly, the risk of COVID-19-related hospitalizations and deaths was 2-fold higher among non-vaccinated South Asians compared to non-vaccinated non-South Asians (aOR 2.03, 95% CI 1.9, 2.2). (Table 3).

## Subgroup analysis by immigration and residential status

Similar to the primary analysis, non-vaccinated South Asians compared to non-vaccinated non-South Asians had a higher odds of COVID-19 infection regardless of their immigration status or reason for immigration (Table 3).

## Subgroup analysis by reason for immigration (economic, refugee, and family/other)

Among immigrants, an increased odds of symptomatic COVID-19 infection were observed among South Asians compared to non-South Asians regardless of the reason for immigration. (Table 3). There was no difference in the risk of COVID-19 related hospitalizations and deaths among non-vaccinated South Asians and non-South Asians based on the reason of immigration, except that South Asians who immigrated to Canada as refugees had a lower risk of COVID-19 related hospitalizations and deaths compared to non-South Asian refugees

**Table 3. Adjusted logistic regression models for various outcomes in non-vaccinated South Asians, stratified by immigration status and reason for immigration (Referent cohort: Non-South Asian non-vaccinated).**

| Overall cohort | | Non-immigrants | | Recent immigrant (<10 years) | | Non-recent immigrant (>10 years) | |
|---|---|---|---|---|---|---|---|
| Covid-19 related hospitalization or death (n = 766464) | Covid-19 infection (n = 883155) | Covid-19 related hospitalization or death (n = 636141) | Covid-19 infection (n = 712271) | Covid-19 related hospitalization or death (n = 38072) | Covid-19 infection (n = 50945) | Covid-19 related hospitalization or death (n = 92251) | Covid-19 infection (n = 119939) |
| 2.03* (1.9, 2.2) | 2.4* (2.3, 2.4) | 2.3* (1.9, 2.6) | 2.5* (2.4, 2.6) | 0.83** (0.6, 1.1) | 1.2* (1.1, 1.2) | 0.99* (0.9, 1.1) | 1.4* (1.3, 1.4) |
| **Total immigrants** | | **Economic** | | **Refugee** | | **Family/others** | |
| Covid-19 related hospitalization or death (n = 130323) | Covid-19 infection (n = 170884) | Covid-19 related hospitalization or death (n = 68501) | Covid-19 infection (n = 84573) | Covid-19 related hospitalization or death (n = 19383) | Covid-19 infection (n = 28041) | Covid-19 related hospitalization or death (n = 42439) | Covid-19 infection (n = 58270) |
| 0.96** (0.9, 1.1) | 1.30* (1.2, 1.3) | 0.99* (0.8, 1.2) | 1.2** (1.2, 1.3) | 0.6* (0.4, 0.9) | 1.1** (1.02, 1.3) | 1.1* (0.9, 1.3) | 1.5** (1.4, 1.6) |

* Hosmer-Lemeshow test for goodness of fit p-value <0.05

** Hosmer-Lemeshow test for goodness of fit p-value >0.05

Adjusted for age, sex, any comorbid condition, rural status, neighborhood income quintile

(Table 3). Similar results were obtained when further stratified analysis by time since and reason for immigration (S8 and S9 Tables).

Similar results were seen when outcomes of COVID-19 related hospitalisations and death were analysed separately (S10 Table) and after excluding those with pre-existing respiratory conditions (S11 Table). We also found similar estimates of COVID-19 vaccine effectiveness based on the stratified analysis by the pandemic waves (S12–S14 Tables).

## Discussion

In this large population-based study including close to 900,000 individuals, we show that in the 2nd to 4th waves of the COVID pandemic in Canada, COVID-19 vaccines were effective in preventing symptomatic SARS CoV-2 infections, hospitalizations and/or deaths among both South Asians and non-South Asians. We also demonstrate that, among non-vaccinated individuals, South Asians had higher odds of COVID-19 infection, and an increased risk of COVID-19 hospitalizations and deaths compared to non-South Asians which was not explained by difference in comorbid, socioeconomic, or immigration-related factors.

The social determinants of health are important to understand the higher impact of COVID-19 among South Asians. Some South Asian communities in Canada are vulnerable groups as they are more likely to live in multigenerational households, earn lower income compared to their education level, and are frequently employed as front-line workers, such as in transit, grocery stores, warehouses, health care, and construction [13]. Due to these reasons, it is imperative to ensure adequate COVID-19 vaccine coverage among South Asians. Data from outside of Canada indicate a substantial amount of misinformation regarding the importance and effectiveness of COVID-19 vaccine in South Asian communities [8–10]. Among South Asians in Canada, the top three sources of information regarding COVID-19 came from health care providers and Public Health officials, national news, and traditional media sources, as well as social media [14]. In addition, outside of Canada, issues such as language barriers, messaging not being tailored for the South Asian community, and lack of access to technology, were observed to be associated with lower rates of vaccine uptake and access [11]. However, in Canada, South Asian advocacy groups and specialized knowledge translation strategies bridged this gap [21]. Finally, ethnic communities including South Asians, were

significantly underrepresented in COVID-19 vaccine trials, leading to sparse data on the effectiveness for this ethnic group [22], which may have affected vaccine confidence and rates of vaccination.

The disproportionate impact of COVID-19 among racialized groups has been confirmed by a number of studies conducted worldwide, [2, 23–27] including recent meta-analyses [26, 28]. In one study which was published prior to the introduction of the COVID-19 vaccines, 18,728,893 patients were included. Black individuals (adjusted RR 2.02, 95% CI 1.67–2.44) and Asians (defined broadly, and differentiation between Chinese and South Asians was not provided) were at a higher risk of COVID-19 infections after adjusting for age, sex, and comorbid conditions (adjusted RR 1.50, 95% CI 1.24–1.83). Moreover, Asians were at higher risk of ICU admission (RR 1.97, 95% CI 1.34–2.89) compared to those of white European descent [28].

Studies evaluating the association between ethnic populations and COVID-19 in Canada are limited due to non-uniform collection of ethnicity data [29]. For that reason, proxy measures of ethnicity are used [1, 3, 4]. In this study, we used a validated last name-based algorithm to identify South Asians living in Ontario. Our findings show that non-vaccinated South Asians have a higher risk of COVID-19 related adverse clinical outcomes. Interestingly, among South Asians, we did not find any differences in the frequency COVID-19 infection among the immigrant and non-immigrant sub-populations. These findings show that there may be several complex sociocultural factors that are in play within the South Asian community which are responsible for their increased risk of COVID-19 related adverse outcomes. It is also plausible that unique biologic pathways such a genetic variation and immune response, among South Asians account for their higher odds of COVID-19 infection. For example, the Angiotensin-Converting Enzyme-2 (ACE2) receptor gene and gene expression affect entry of SARS COV-2 into cells and varies between ethnic groups and may be implicated in differential susceptibility and severity to SARS CoV-2 among South Asians [30, 31]. Additionally, the potential impact of HLA and genetic factors on COVID-19 risk among different ancestral groups is also an important area for further research [32–34], and more such studies are needed among South Asians to understand their increased COVID-19 susceptibility and severity.

To our knowledge, this is the first study to evaluate the effectiveness of COVID-19 vaccines among South Asians using a test-negative design at a population level. This study design mitigates potential bias arising from differences in access to healthcare by restricting to those individuals who presented for SARS-CoV-2 testing. Despite our study's strengths, there are some limitations that deserve mention. First, our results are generalizable only to those healthcare systems with universal healthcare coverage like Ontario. Second, there is a possibility those individuals with no information on symptoms recorded in the databases might have had symptoms at the time of testing, and those recorded as asymptomatic might have subsequently developed symptoms. Third, the last name-based algorithm only allows us to identify South Asian and Chinese ethnic groups but not other ethnic communities, thus the non-South Asian group was heterogeneous by ethnicity. Fourth, we did not have data on employment, education, gender, and other social factors to further understand the increased risk of adverse outcomes among non-vaccinated South Asians. Finally, due to administrative nature of our databases, there is a potential for residual confounding and measurement error for a few variables.

## Conclusions

This study shows that COVID-19 vaccines are effective in reducing COVID-19 infections, hospitalizations and death among South Asians living in Ontario, Canada. These results

should provide reassurance and foster confidence among the South Asian community regarding vaccination. Non-vaccinated South Asians have a higher odds of COVID-19 related adverse outcomes compared to non-South Asians. Future studies are needed to explain the higher risk of COVID-19 infection and worse outcomes among non-vaccinated South Asians.

## Supporting information

**S1 Table. List of covariates and their codes used in the study.**
(DOCX)

**S2 Table. Cohort creation process.**
(DOCX)

**S3 Table. Baseline characteristics of the overall cohort stratified by covid-19 test results.**
(DOCX)

**S4 Table. Baseline comorbidities of the overall cohort stratified by ethnicity and vaccination status.**
(DOCX)

**S5 Table. Baseline comorbidities of the overall cohort stratified by covid-19 test results.**
(DOCX)

**S6 Table. Vaccine effectiveness among South Asians and non-South Asians when COVID-19 associated hospitalization and deaths were analysed separately.**
(DOCX)

**S7 Table. Vaccine effectiveness among South Asians and non-South Asians after excluding those with pre-existing respiratory conditions.**
(DOCX)

**S8 Table. Adjusted logistic regression models for symptomatic covid-19 infection in non-vaccinated South Asians, stratified by reason of immigration and years (Referent cohort: Non-South Asian non-vaccinated).**
(DOCX)

**S9 Table. Adjusted logistic regression models for covid-19 related hospitalization or death, stratified by type of immigration and years in non-vaccinated South Asians (Referent cohort: Non-South Asian non-vaccinated).**
(DOCX)

**S10 Table. Adjusted logistic regression models for outcomes of COVID-19 related hospitalizations and death analyzed separately in non-vaccinated South Asians, stratified by immigration status and reason for immigration (Referent cohort: Non-South Asian non-vaccinated).**
(DOCX)

**S11 Table. Adjusted logistic regression models for various outcomes of COVID-19 in non-vaccinated South Asians, stratified by immigration status and reason for immigration (Referent cohort: Non-South Asian non-vaccinated) after excluding those with pre-existing respiratory conditions.**
(DOCX)

**S12 Table. Vaccine effectiveness among South Asians and non-South Asians in Wave 2 of COVID-19 pandemic [(Dec 14 2020 (start of our data) to 28 February 2021].**
(DOCX)

**S13 Table. Vaccine effectiveness among South Asians and non-South Asians in Wave 3 of COVID-19 pandemic (March 1 2021 to July 31 2021).**
(DOCX)

**S14 Table. Vaccine effectiveness among South Asians and non-South Asians in Wave 4 of COVID-19 pandemic (Wave 4: August 1 2021 to November 15, 2021 (end of our data).**
(DOCX)

**S1 Fig. Participant flow diagram.**
(DOCX)

## Acknowledgments

We acknowledge the support and guidance we received from the entire ICES team. Parts of this material are based on data and information compiled and provided by MOHLTC and CIHI. This work was also supported by the Ontario Health Data Platform (OHDP), a Province of Ontario initiative to support Ontario's ongoing response to COVID-19 and its related impacts. This document used data adapted from the Statistics Canada Postal Code^OM Conversion File, which is based on data licensed from Canada Post Corporation, and/or data adapted from the Ontario Ministry of Health Postal Code Conversion File, which contains data copied under license from Canada Post Corporation and Statistics Canada. Parts of this material are based on data and information compiled and provided by the MOH, the Canadian Institute for Health Information (CIHI), and Immigration Refugees and Citizenship Canada. The analyses, conclusions, opinions and statements expressed herein are solely those of the authors and do not reflect those of the funding or data sources; no endorsement is intended or should be inferred. No endorsement by the OHDP, its partners, or the Province of Ontario is intended or should be inferred. The analyses, conclusions, opinions and statements expressed herein are solely those of the authors and do not reflect those of the funding or data sources; no endorsement is intended or should be inferred.

## Author Contributions

**Conceptualization:** Rahul Chanchlani, Shrikant I. Bangdiwala, Russell J. de Souza, Shelly Bolotin, Dawn M. E. Bowdish, Dipika Desai, Scott A. Lear, Mark Loeb, Zubin Punthakee, Diana Sherifali, Gita Wahi, Sonia S. Anand.

**Data curation:** Baiju R. Shah.

**Formal analysis:** Rahul Chanchlani, Baiju R. Shah, Russell J. de Souza, Jin Luo, Karl Everett, Mark Loeb.

**Investigation:** Rahul Chanchlani, Shelly Bolotin, Sonia S. Anand.

**Methodology:** Rahul Chanchlani, Shrikant I. Bangdiwala.

**Supervision:** Sonia S. Anand.

**Writing – original draft:** Rahul Chanchlani, Sonia S. Anand.

**Writing – review & editing:** Baiju R. Shah, Shrikant I. Bangdiwala, Russell J. de Souza, Jin
Luo, Shelly Bolotin, Dawn M. E. Bowdish, Dipika Desai, Karl Everett, Scott A. Lear, Mark
Loeb, Zubin Punthakee, Diana Sherifali, Gita Wahi.

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
