## [Decision Letter · Decision Letter 0]

16 Feb 2024

PGPH-D-23-02557

COVID-19 vaccine effectiveness among South Asians in Canada

Dear Dr. Anand,

Thank you for submitting your manuscript to PLOS Global Public Health. After careful consideration, we feel that it has merit but does not fully meet PLOS Global Public Health’s publication criteria as it currently stands. Therefore, we invite you to submit a revised version of the manuscript that addresses the points raised during the review process.

We look forward to receiving your revised manuscript.

Kind regards,

Jianhong Zhou

Staff Editor

Journal Requirements:

Additional Editor Comments (if provided):

Reviewers' comments:

Reviewer's Responses to Questions

**Comments to the Author**

1. Does this manuscript meet PLOS Global Public Health’s publication criteria? Is the manuscript technically sound, and do the data support the conclusions? The manuscript must describe methodologically and ethically rigorous research with conclusions that are appropriately drawn based on the data presented.

Reviewer #1: Yes

Reviewer #2: Yes

2. Has the statistical analysis been performed appropriately and rigorously?

Reviewer #1: Yes

Reviewer #2: Yes

3. Have the authors made all data underlying the findings in their manuscript fully available (please refer to the Data Availability Statement at the start of the manuscript PDF file)?

Reviewer #1: Yes

Reviewer #2: Yes

4. Is the manuscript presented in an intelligible fashion and written in standard English?

Reviewer #1: Yes

Reviewer #2: Yes

5. Review Comments to the Author

Reviewer #1: I do have some comments/feedback:

Line 61-62: Was there an IRB non-research determination?

Line 78: Why one at random in determining the index and not the first one? Why not the CSTE case definition as is used in the literature. DO you have a justification for using this? If not, I recommend using the CSTE case definition including the 90 day rule for reinfections found in a the following site.

https://ndc.services.cdc.gov/case-definitions/coronavirus-disease-2019-2021/

Lines 82-83: Why not the CDC recommended definition for fully vaccinated which is 14 days after the 2nd dose? Do you have a justification for this? If not, I recommend using the CDC recommended definition used in the literature. At a minimum please add a sensitivity analysis using the 14 day definition.

Lines 104-105: This looks like the CSTE definition for hospitalizations. I suggest referencing that.

Tables 1 and 2: Is this p value for a group chi-square test? IF so can you specify?

Comments on statistical analysis:

I suggest adding overall model goodness of fit statistics associated with tables 3 and 4.

Close to 30% of the total cases and controls we excluded due to various reasons including due to missing values. Not sure what % of that is <18 years. For others, recommend conducting a sensitivity analysis or at a minimum discuss the potential impact/limitations.

Comment on data availability:

Whether the files made available is the matched analysis file or the original data is not clear. If possible, suggest making the matched analysis file available.

Reviewer #2: This study evaluates the effectiveness of COVID-19 vaccines among South Asians compared to non-South Asians in Ontario, Canada, using a test-negative study design. It reveals a high COVID-19 vaccine effectiveness among South Asians in Ontario and indicates that non-vaccinated South Asians have higher odds of symptomatic COVID-19 infection. Overall, the study is well-written and informative, but I have several suggestions, particularly regarding the statistical methods employed, which I believe could enhance the paper:

• Have hospitalizations and deaths been investigated separately? I could imagine that hospitalizations drive a bulk of the results and would be a relevant sensitivity analysis.

• Line 66: Clarification is needed on how symptoms consistent with COVID-19 are defined. Did you use certain diagnostic codes or is it based on a PCR test?

• Table 2 & Line 123: Are all comorbidities listed in Table 2 combined in the model as one covariate for any comorbidity? If so, consider incorporating a score or grouping comorbidities (e.g., respiratory, cardiovascular) for a more comprehensive analysis. Having an extensive Table 2 with all the comorbidities represented and then not being as detailed with the covariate in the analysis seems excessive. Moving Table 2 to the supplementary section and presenting the score in Table 1 could improve clarity. Additionally, conducting sensitivity analyses, such as excluding individuals with preexisting respiratory conditions or stratifying by them, could be informative. One could hypothesize that having preexisting conditions related to COVID could make you more susceptible.

• Line 128: Missing hyphen in non-vaccinated.

• Line 136: Including 126,016 and 757,139 in the flowchart for completeness would enhance clarity.

• Lines 145-146: Immigrant percentages mentioned are not found in any of the tables.

• Table 4: I find table 4 confusing, especially the fact that it has two ‘Overall’ categories. I would recommend having more informative titles.

• Logistic regression: The use of logistic regression may overlook the time-varying nature of the COVID-19 pandemic and assumes a static relationship between predictors and outcomes. Given the influence of temporal factors such as COVID infections, variants, seasonal variations, and vaccination roll-outs, Cox regression or conditional logistic regression, would be a more suitable approach. The conditional logistic regression could potentially be matched on time. VE could be confounded by changes in the underlying risk of COVID-19 infection over time, leading to biased results. An initial check to see if there is any time-varying confounding is to conduct a sensitivity analysis to see if stratifying by time periods has an effect on your estimates.

• Given the variant variation and the fluctuating effectiveness of vaccines dependent on variants, could you provide a short description of the variant dominance in Canada during the study period? The timing of vaccination could also have an effect on the effectiveness in this year-long follow-up. Consider including this covariate as an adjuster in your analysis.

6. PLOS authors have the option to publish the peer review history of their article (what does this mean?). If published, this will include your full peer review and any attached files.

**Do you want your identity to be public for this peer review?** For information about this choice, including consent withdrawal, please see our Privacy Policy.

Reviewer #1: **Yes: **Vajeera Dorabawila, PhD

Reviewer #2: No

---

## [Decision Letter · Decision Letter 1]

16 May 2024

PGPH-D-23-02557R1

COVID-19 vaccine effectiveness among South Asians in Canada

Dear Dr. Anand,

Thank you for submitting your manuscript to PLOS Global Public Health. After careful consideration, we feel that it has merit but does not fully meet PLOS Global Public Health’s publication criteria as it currently stands. Therefore, we invite you to submit a revised version of the manuscript that addresses the points raised during the review process.

The revised manuscript has been re-evaluated by two reviewers, and their comments are available below. Both reviewers are happy with the revisions to the manuscript and responses to their comments. Reviewer 2 has suggested the use of a sensitivity analysis to address any issues with temporal timing and some minor revisions to improve the manuscript.

We look forward to receiving your revised manuscript.

Kind regards,

Emma Campbell, Ph.D

Staff Editor

Journal Requirements:

1. Please amend your detailed online Financial Disclosure statement. This is published with the article. It must therefore be completed in full sentences and contain the exact wording you wish to be published.

a) State the initials, alongside each funding source, of each author to receive each grant, if applicable. For example: "This work was supported by the National Institutes of Health (####### to AM; ###### to CJ) and the National Science Foundation (###### to AM)."

2. Please ensure that the funders and grant numbers match between the Financial Disclosure field and the Funding Information tab in your submission form. Note that the funders must be provided in the same order in both places as well.

Reviewers' comments:

Reviewer's Responses to Questions

**Comments to the Author**

1. If the authors have adequately addressed your comments raised in a previous round of review and you feel that this manuscript is now acceptable for publication, you may indicate that here to bypass the “Comments to the Author” section, enter your conflict of interest statement in the “Confidential to Editor” section, and submit your "Accept" recommendation.

Reviewer #1: All comments have been addressed

Reviewer #2: (No Response)

2. Does this manuscript meet PLOS Global Public Health’s publication criteria? Is the manuscript technically sound, and do the data support the conclusions? The manuscript must describe methodologically and ethically rigorous research with conclusions that are appropriately drawn based on the data presented.

Reviewer #1: Yes

Reviewer #2: (No Response)

3. Has the statistical analysis been performed appropriately and rigorously?

Reviewer #1: Yes

Reviewer #2: (No Response)

4. Have the authors made all data underlying the findings in their manuscript fully available (please refer to the Data Availability Statement at the start of the manuscript PDF file)?

Reviewer #1: Yes

Reviewer #2: (No Response)

5. Is the manuscript presented in an intelligible fashion and written in standard English?

Reviewer #1: Yes

Reviewer #2: Yes

6. Review Comments to the Author

Reviewer #1: na

Reviewer #2: Line 119: Please specify the 4 groups

Line 139: “A total of 883,155 individuals who had PCR test” -> “A total of 883,155 individuals who had a PCR test”

Table 3: The footnotes in table 3 look like exponents. I would recommend changing this notation. You could just include the p-value output in the table and add row titles or add the footnote to the subtitle.

Line 200: isn’t the 2nd and 4th wave also included in the study period?

I still think there is a potential issue with temporal timing that is not addressed. During the study period, COVID came in separate waves – some perhaps more severe than others. If most individuals get vaccinated towards the middle/end of the study, then the individuals who are in the vaccinated group most likely only consist of the last wave in the study period. Furthermore, the unvaccinated would then span over the whole period while the vaccinated most likely includes those who test positive towards the end of the period, giving an uneven opportunity for a positive test throughout the study, potentially biasing the result towards more effect. As a sensitivity analysis, this could be checked by stratifying the analysis by the study period (tests before/after June, for example), or otherwise limiting your study period to a time when most individuals have been vaccinated (e.g., > June). By limiting it to a time period where vaccination coverage is more uniform, you can check if the effect of vaccination is constant over time periods. Alternatively, the methods paper you refer to (15) stratifies by the following to take into account the effect of waves and variants: “epidemic wave (index dates 14 December 2020 to 7 February 2021, representing wave 2 in Ontario; 8 February 2021 to 21 March 2021, representing the period between wave 2 and wave 3; and 22 March 2021 to 19 April 2021, representing wave 3), and variant (earlier variant versus alpha versus beta or gamma).”

7. PLOS authors have the option to publish the peer review history of their article (what does this mean?). If published, this will include your full peer review and any attached files.

**Do you want your identity to be public for this peer review?** For information about this choice, including consent withdrawal, please see our Privacy Policy.

Reviewer #1: No

Reviewer #2: No

---

## [Decision Letter · Decision Letter 2]

25 Jun 2024

COVID-19 vaccine effectiveness among South Asians in Canada

PGPH-D-23-02557R2

Dear Dr. Anand,

We are pleased to inform you that your manuscript 'COVID-19 vaccine effectiveness among South Asians in Canada' has been provisionally accepted for publication in PLOS Global Public Health.

Best regards,

Julia Robinson

Executive Editor

Reviewer Comments (if any, and for reference):

Reviewer's Responses to Questions

**Comments to the Author**

1. If the authors have adequately addressed your comments raised in a previous round of review and you feel that this manuscript is now acceptable for publication, you may indicate that here to bypass the “Comments to the Author” section, enter your conflict of interest statement in the “Confidential to Editor” section, and submit your "Accept" recommendation.

Reviewer #1: All comments have been addressed

Reviewer #2: All comments have been addressed

2. Does this manuscript meet PLOS Global Public Health’s publication criteria? Is the manuscript technically sound, and do the data support the conclusions? The manuscript must describe methodologically and ethically rigorous research with conclusions that are appropriately drawn based on the data presented.

Reviewer #1: Yes

Reviewer #2: (No Response)

3. Has the statistical analysis been performed appropriately and rigorously?

Reviewer #1: Yes

Reviewer #2: (No Response)

4. Have the authors made all data underlying the findings in their manuscript fully available (please refer to the Data Availability Statement at the start of the manuscript PDF file)?

Reviewer #1: Yes

Reviewer #2: (No Response)

5. Is the manuscript presented in an intelligible fashion and written in standard English?

Reviewer #1: Yes

Reviewer #2: (No Response)

6. Review Comments to the Author

Reviewer #1: The second round comments were from the other reviewer. Thank you for sharing the revised version.

Reviewer #2: (No Response)

7. PLOS authors have the option to publish the peer review history of their article (what does this mean?). If published, this will include your full peer review and any attached files.

**Do you want your identity to be public for this peer review?** For information about this choice, including consent withdrawal, please see our Privacy Policy.

Reviewer #1: No

Reviewer #2: No
